# Identifiability and numerical algebraic geometry

**Daniel J. Bates**[1☯], **Jonathan D. Hauenstein**[2☯]*, **Nicolette Meshkat**[3☯]

**1** Department of Mathematics, United States Naval Academy, Annapolis, MD, United States of America,
**2** Department of Applied and Computational Mathematics and Statistics, University of Notre Dame, Notre Dame, IN, United States of America, **3** Department of Mathematics and Computer Science, Santa Clara University, Santa Clara, CA, United States of America

☯ These authors contributed equally to this work.
* hauenstein@nd.edu

## Abstract

A common problem when analyzing models, such as mathematical modeling of a biological process, is to determine if the unknown parameters of the model can be determined from given input-output data. Identifiable models are models such that the unknown parameters can be determined to have a finite number of values given input-output data. The total number of such values over the complex numbers is called the identifiability degree of the model. Unidentifiable models are models such that the unknown parameters can have an infinite number of values given input-output data. For unidentifiable models, a set of identifiable functions of the parameters are sought so that the model can be reparametrized in terms of these functions yielding an identifiable model. In this work, we use numerical algebraic geometry to determine if a model given by polynomial or rational ordinary differential equations is identifiable or unidentifiable. For identifiable models, we present a novel approach to compute the identifiability degree. For unidentifiable models, we present a novel numerical differential algebra technique aimed at computing a set of algebraically independent identifiable functions. Several examples are used to demonstrate the new techniques.

**Data Availability Statement:** All data files are publicly available at http://dx.doi.org/10.7274/R03T9F91.

**Funding:** DJB was supported by National Science Foundation (www.nsf.gov) ACI-1440467 and National Science Foundation (www.nsf.gov) DMS-

## Introduction

Parameter identifiability analysis for dynamical system models consisting of ordinary differential equations (ODEs) addresses the question of which unknown parameters can be determined from given input-output data. In this paper, we address *structural* identifiability, which concerns whether the parameters of a model can be determined from *perfect* input-output data, i.e., noise-free and of any time duration required. This is a necessary condition for the *practical* or *numerical* identifiability problem, which involves parameter estimation with real, and often noisy, data. For this reason, structural identifiability is often referred to as *a priori* identifiability [1]. Even if a model fails to be structurally identifiable, some useful information about the parameters can still be determined, which is the main motivation for this paper.

1719658. JDH was supported by Army (https://www.arl.army.mil/) YIP W911NF-15-1-0219, National Science Foundation (www.nsf.gov) ACI-1460032, and Sloan Research Fellowship (https://sloan.org/fellowships/) BR2014-110 TR14. NM was supported by Luce Foundation (https://www.hluce.org/). The funders had no role in study design, data collection and analysis, decision to publish, or preparation of the manuscript.

**Competing interests:** The authors have declared that no competing interests exist.

There are two possible outcomes of the structural identifiability check of a mathematical model. If the parameters of the model have a unique or finite number of values given input-output data, then the model and its parameters are said to be *identifiable*. However, if some subset of the parameters can take on an infinite number of values and yet yield the same input-output data, then the model and this subset of parameters are called *unidentifiable*. In the latter case, we attempt to find a set of *identifiable functions* of the parameters. These can then be used to reparameterize the model and also to give additional insight into which parameters should be experimentally measured [2].

Several methods have been proposed to find identifiable functions. In linear models, this can be done using the transfer function method [3]. However, in nonlinear models, the problem has been more challenging with only *ad hoc* methods proposed, e.g., [2, 4, 5]. For example, the approach in [2] requires the calculation of many Gröbner bases and can thus be computationally expensive. It should be noted, however, that even in the linear case, the identifiable functions of parameters found using the transfer function method are not necessarily (and are usually not) the *simplest* identifiable functions of parameters. Since our goal is to reparametrize a model over identifiable functions of the parameters, simpler functions are preferred.

In this paper, we use techniques from numerical algebraic geometry (e.g., see [6, 7] for a general overview) to investigate both identifiable and unidentifiable models. For an identifiable model, we compute the finite number of values of the parameters given input-output data. The total number of such values over the complex numbers is called the *identifiability degree* which is computed in two ways. The first method relies on differential algebra tools to first generate the *input-output equations* while the second does not utilize these equations.

For unidentifiable models, we also introduce two novel approaches for finding identifiable functions of the parameters. The first method relies on knowing the *input-output equations* and uses them to find *globally identifiable* functions of parameters, as in [2]. In the case where these input-output equations cannot be calculated using conventional differential algebra techniques, we also introduce a method to compute *locally identifiable* functions of parameters. This combination of numerical algebraic geometry and differential algebra could be thought of as *numerical differential algebra*. We demonstrate our methods on various models.

## Materials and methods

### Identifiability

We consider ODE models of the form:

$$
\begin{aligned}
\dot{\mathbf{x}}(t) &= \mathbf{f}(\mathbf{x}(t), \mathbf{p}, \mathbf{u}(t), t) \\
\mathbf{y}(t) &= \mathbf{g}(\mathbf{x}(t), \mathbf{p}, t)
\end{aligned}
\tag{1}
$$

where $\mathbf{f}$ and $\mathbf{g}$ are vectors of rational functions, $\mathbf{x}(t)$ is the *state variable* vector, $\mathbf{p}$ is the *parameter* vector which is assumed to be constant, $\mathbf{u}(t)$ is the *input* vector, and $\mathbf{y}(t)$ is the *output* vector. In the following, only the input $\mathbf{u}(t)$ and output $\mathbf{y}(t)$ vectors are assumed to be known, i.e., the state variables $\mathbf{x}(t)$ and the parameters $\mathbf{p}$ are unknown.

**Input-output equations.** One approach to determine identifiability properties of the model (1) using known input-output data is via the *input-output equations*, i.e., equations that relate the input $\mathbf{u}(t)$, output $\mathbf{y}(t)$, and parameters $\mathbf{p}$. Thus, the input-output equations result from eliminating the state variables $\mathbf{x}(t)$. Several methods have already been proposed, e.g., [5, 8–16], to compute the input-output equations, including the so-called *differential algebra approach* to identifiability [11, 13, 15]. Using differential algebra, the state variables $\mathbf{x}(t)$ are eliminated using differential elimination techniques. If the number of outputs $\mathbf{y}(t)$ is $m$, this

procedure produces $m$ differential polynomial equations that are solely in input and output variables with rational coefficients in the parameters so that the $j^{\text{th}}$ one can be written as

$$\sum_i c_{ji}(\mathbf{p})\psi_i(\mathbf{u}, \mathbf{y}) = 0 \tag{2}$$

where each $\psi_i(\mathbf{u}, \mathbf{y})$ is a differential monomial. Each $c_{ji}(\mathbf{p})$ is a rational function in the parameters $\mathbf{p}$, forming a collection $\mathbf{c}(\mathbf{p})$ called the *coefficients of the input-output equations*. The coefficients of each input-output equation can be determined uniquely by normalizing each input-output equation so that one of the coefficients is one.

**Deciding identifiability.** Let $m_1$ denote the number of independent parameters $\mathbf{p}$ and $m_2$ denote the total number of non-constant coefficients taken from all $m$ input-output equations. Thus, we can treat the coefficients of the input-output equations as a rational map $\mathbf{c} : \mathbb{C}^{m_1} \to \mathbb{C}^{m_2}$. Identifiability refers to whether it is possible to recover the parameters of the model only by observing the relations among the input and output variables. In other words, assuming known input-output data for a sufficient number of time instances so that $\mathbf{c}$ can theoretically be computed, identfiability asks whether it is possible to recover the parameters $\mathbf{p}$.

**Definition 1**. Let $\mathbf{c}$ be the coefficients of the input-output equations for a model (1). For general $\mathbf{p} \in \mathbb{C}^{m_1}$, let

$$X_{\mathbf{p}} = \mathbf{c}^{-1}(\mathbf{c}(\mathbf{p})) = \{\mathbf{q} \in \mathbb{C}^{m_1} \mid \mathbf{c}(\mathbf{q}) = \mathbf{c}(\mathbf{p})\} \subset \mathbb{C}^{m_1}, \tag{2}$$

$\ell = \dim X_{\mathbf{p}} \geq 0$, and $k = \#X_{\mathbf{p}} \in \mathbb{N} \cup \{\infty\}$. That is, $\ell$ is the dimension of a general fiber of $\mathbf{c}$ and $\mathbf{c}$ is generically a $k$-to-one map when $\ell = 0$. The model (1) is *identifiable* from $\mathbf{c}$ if $\ell = 0$, i.e., $k \in \mathbb{N}$, and *unidentifiable* if $\ell > 0$, i.e., $k = \infty$.

When identifiable, the number $k \in \mathbb{N}$ is called the *identifiability degree*. If $k = 1$, the model (1) is called *globally identifiable* and called *locally identifiable* if $1 < k < \infty$.

When unidentifiable, the number $\ell \geq 1$ is called the *dimension of unidentifiability*.

To distinguish between identifiable and unidentifiable models, one simply needs to compute the dimension $\ell$ of a general fiber of $\mathbf{c}$. As defined in Section 13.4 of [7], the *rank* of $\mathbf{c}$, denoted rank $\mathbf{c}$, is the rank of the Jacobian matrix of $\mathbf{c}$ evaluated at a general, i.e., random, $\mathbf{p} \in \mathbb{C}^{m_1}$. The *corank* of $\mathbf{c}$ is corank $\mathbf{c} = m_1 -$ rank $\mathbf{c}$. The following, which is Lemma 13.4.1 of [7] (see also [17]), relates $\ell$ and corank $\mathbf{c}$.

**Proposition 2**. *For a general* $\boldsymbol{p} \in \mathbb{C}^{m_1}$, $\ell = \dim X_{\boldsymbol{p}}$ *as defined in* (2) *is equal to* corank $\boldsymbol{c}$ *where* $\boldsymbol{c}$ *is the set of coefficients of the input-output equations. In particular, the model* (1) *is identifiable if and only if* $\boldsymbol{c}$ *has full rank and the dimension of unidentifiability is equal to* corank $\boldsymbol{c}$.

In particular, Prop. 2 indicates a method to distinguish between identifiable and unidentifiable models provided that the coefficients $\mathbf{c}$ of the input-output equations can be computed, which is summarized in the following pseudocode.

**Method 1:** Computing dimension of unidentifiability from input-output equations
**Input:** $m_2$ input-output equation coefficients `c(p)`, depending on parameters $\mathbf{p} = (p_1, \dots, p_{m_1})$.
**Output:** Dimension of unidentifibility $\ell$ = corank `c` = dim $\mathbf{c}^{-1}(\mathbf{c}(\mathbf{q}))$ for general $\mathbf{q} \in \mathbb{C}^{m_1}$.
Choose random, complex values $\mathbf{q} \in \mathbb{C}^{m_1}$.
Return $\ell$ = corank $J_{\mathbf{c}}(\mathbf{q})$ where $J\mathbf{c}(\mathbf{p})$ is the Jacobian matrix of `c` evaluated at $\mathbf{p}$.

**Example 3**. Linear compartment models are frequently used models arising in pharmacokinetics, toxicology, cell biology, physiology, and ecology [18–22]. The following from [17] is an example of a linear three-compartment model with input $u(t)$, output $y(t)$, state variables $\mathbf{x}(t) = (x_1(t), x_2(t), x_3(t))$, and unknown parameters $\mathbf{p} = (k_{01}, k_{02}, k_{03}, k_{12}, k_{13}, k_{21}, k_{32})$, where $k_{ij}$

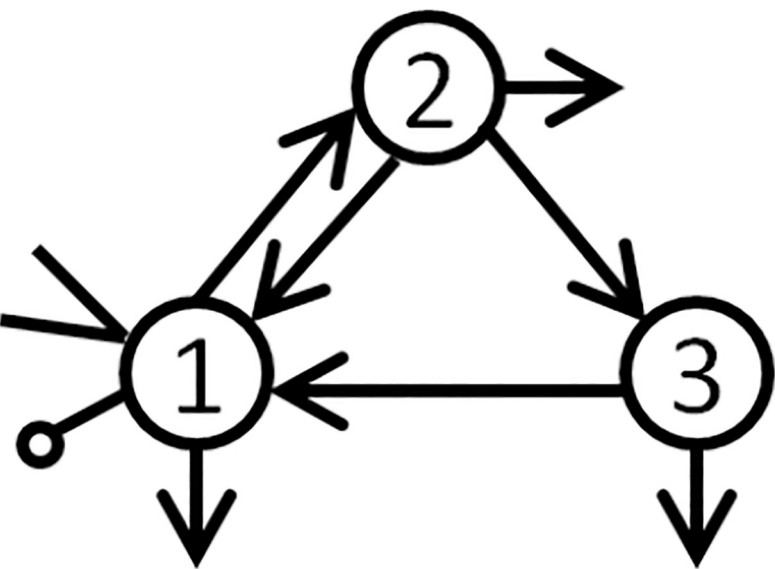

**Fig 1. A 3-compartment model.** A 3-compartment model with input (represented by the arrowhead) and output (represented by the line segment with a circle at the end) in the first compartment and "leaks" from every compartment (represented by arrows leaving the compartments). Here, the input could represent a drug injection and the first compartment could represent blood, with the other two compartments representing organs, e.g., tissue and stomach. The unknown parameters represent rates of transfer from one compartment to another (drawn as arrows in the figure), or in the case of leaks, from one compartment to outside the system. The state variables represent drug concentration in the blood and organs, with output from the first compartment representing measured drug concentration in the blood.

represents the rate of transfer from compartment $j$ to compartment $i$ and $k_{0i}$ represents a leak from compartment $i$ to outside the system:

$$
\begin{aligned}
\dot{x}_1 &= -(k_{01} + k_{21})x_1 + k_{12}x_2 + k_{13}x_3 + u \\
\dot{x}_2 &= k_{21}x_1 - (k_{02} + k_{12} + k_{32})x_2 \\
\dot{x}_3 &= k_{32}x_2 - (k_{03} + k_{13})x_3 \\
y &= x_1.
\end{aligned}
\tag{3}
$$

Fig 1 presents a pictorial representation of this model.

The approach described in [17, 23] yields the input-output equation:

$$
\dddot{y} - c_1(\mathbf{p})\ddot{y} + c_2(\mathbf{p})\dot{y} - c_3(\mathbf{p})y - \ddot{u} + c_4(\mathbf{p})\dot{u} - c_5(\mathbf{p})u = 0
$$

where

$$
\begin{aligned}
c_1(\mathbf{p}) &= E_1(-(k_{01} + k_{21}), -(k_{02} + k_{12} + k_{32}), -(k_{03} + k_{13})) \\
c_2(\mathbf{p}) &= E_2(-(k_{01} + k_{21}), -(k_{02} + k_{12} + k_{32}), -(k_{03} + k_{13})) - E_2(k_{12}, k_{21}) \\
c_3(\mathbf{p}) &= E_3(-(k_{01} + k_{21}), -(k_{02} + k_{12} + k_{32}), -(k_{03} + k_{13})) + E_3(k_{13}, k_{32}, k_{21}) \\
&\quad + E_3(k_{12}, k_{21}, (k_{03} + k_{13})) \\
c_4(\mathbf{p}) &= E_1(-(k_{02} + k_{12} + k_{32}), -(k_{03} + k_{13})) \\
c_5(\mathbf{p}) &= E_2(-(k_{02} + k_{12} + k_{32}), -(k_{03} + k_{13}))
\end{aligned}
$$

such that $E_k(z_1, \ldots, z_m)$ is the $k^{\text{th}}$ elementary symmetric polynomial in $z_1, \ldots, z_m$. Thus, for $\mathbf{c} = (c_1, \ldots, c_5)$, it is easy to verify that rank $\mathbf{c} = 5$ and $\mathbf{c} = 2$ so that this model is unidentifiable with 2 dimensions of unidentifiability.

For an identifiable model, one approach to distinguish between global and local identifiability is to solve the system of equations $\mathbf{c}(\mathbf{q}) = \mathbf{c}(\mathbf{p})$ given a general point $\mathbf{p} \in \mathbb{C}^{m_1}$. If there is a unique solution, namely $\mathbf{q} = \mathbf{p}$, the model is globally identifiable. If there are a finite number of solutions, the model is locally identifiable. Such an approach, for example, is implemented in the software package DAISY [1, 24] which randomly selects a point $\mathbf{p}$ and uses Gröbner basis methods to count the number of solutions to $\mathbf{c}(\mathbf{q}) = \mathbf{c}(\mathbf{p})$ yielding the identifiability degree. Since such an approach can only be applied when $\mathbf{c}$ has first been computed, we will consider the following problem using numerical algebraic geometric methods.

**Problem 4**. Given a model (1), decide if it is identifiable or unidentifiable. If identifiable, determine its identifiability degree to decide if it is globally identifiable or locally identifiable.

One technique for determining whether a model is identifiable without computing $\mathbf{c}$ is via the Exact Arithmetic Rank (EAR) approach [25]. In particular, rather than eliminating to compute the corank of $\mathbf{c}$, one considers projections of a system that still involves the state variables derived by replacing functions with Taylor series expansions and taking a finite-size system via the Cartan-Kuranishi Theorem that underlies differential elimination, e.g., see [26]. Projections yield contructible algebraic sets whose closure in both the Euclidean and Zariksi topologies are equal. The following, Lemma 3 from [27], is essential for computing corank $\mathbf{c}$ without first computing $\mathbf{c}$.

**Proposition 5**. *Let $F : \mathbb{C}^N \to \mathbb{C}^n$ be a polynomial system, $V \subset \{x \in \mathbb{C}^N \mid F(x) = 0\} \subset \mathbb{C}^N$ be irreducible of multiplicity* 1 *with respect to F, and $\pi(x_1, \ldots, x_N) = (x_1, \ldots, x_a)$ for some $a \leq N$. For general $z \in V$,*

$$\dim \overline{\pi(V)} = \text{corank}_0 \, JF(z) - \text{corank}_a \, JF(z)$$

*where $JF(z)$ is the Jacobian matrix of F evaluated at z and $\text{corank}_j \, M$ is the corank of the last $N - j$ columns of M.*

**Example 6**. With the setup from Ex. 3, write the function $\mathbf{x}(t)$, $u(t)$, and $y(t)$ using a Taylor series expansion centered at $t = 0$, namely

$$\mathbf{x}(t) = \sum_{j=0}^{\infty} \mathbf{x}_j \cdot t^j/j!, \quad u(t) = \sum_{j=0}^{\infty} u_j \cdot t^j/j!, \quad \text{and} \quad y(t) = \sum_{j=0}^{\infty} y_j \cdot t^j/j!. \tag{4}$$

Since (3) holds for all $t$, one obtains equations by substituting these Taylor series expansions into (3) and taking coefficients with respect to $t$. For $r \geq 0$, let $F_r$ be the system obtained by taking coefficients of $1, t, t^2, \ldots, t^r$. For this linear compartment model, the coefficients of $t^j$ are

$$G_j = \begin{bmatrix} -(k_{01} + k_{21})x_{j1} + k_{12}x_{j2} + k_{13}x_{j3} + u_j - x_{j+1,1} \\ k_{21}x_{j1} - (k_{02} + k_{12} + k_{32})x_{j2} - x_{j+1,2} \\ k_{32}x_{j2} - (k_{03} + k_{13})x_{j3} - x_{j+1,3} \\ y_j - x_{j1} \end{bmatrix} \quad \text{so that} \quad F_r = \begin{bmatrix} G_0 \\ G_1 \\ \vdots \\ G_r \end{bmatrix}.$$

Based on the structure of each $G_j$, it is clear that the Jacobian matrix of $F_r$ has full rank, namely $4(r + 1)$, at every point. In fact, $F_r = 0$ defines an irreducible and smooth solution set of codimension $4(r + 1)$ (dim = $11 + r$). We can compute a random point on this solution set by

randomly selecting the following $11 + r$ values: $\mathbf{p}$, $\mathbf{x}_0$, and $u_0, \ldots, u_r$, and trivially computing the unique $\mathbf{x}_{j+1}$ and $y_j$ sequentially for $j = 0, \ldots, r$ via $G_j = 0$.

Next, one treats the coefficients of the input $u(t)$ and output $y(t)$ as constants in $F_r$. Thus, we have that $F_r$ depends upon $N_r = 13 + 3r$ variables and apply Prop. 5 to compute

$$d_r = \operatorname{corank}_0 JF_r(\mathbf{p}, \mathbf{x}_0, \ldots, \mathbf{x}_{r+1}) - \operatorname{corank}_7 JF_r(\mathbf{p}, \mathbf{x}_0, \ldots, \mathbf{x}_{r+1})$$

since $\pi_r(\mathbf{p}, \mathbf{x}_0, \ldots, \mathbf{x}_{r+1}) = \mathbf{p} \in \mathbb{C}^7$. We trivially know $d_r \geq d_{r+1}$ since $F_r$ is a subset of $F_{r+1}$. Hence, $\{d_r\}_{r=0}^{\infty}$ is a sequence of nonincreasing nonnegative integers that stabilizes with

$$\lim_{r \to \infty} d_r = \operatorname{corank} \mathbf{c}.$$

This limit is obtained at a finite value of $r$ in accordance with the Cartan-Kuranishi Theorem and can be observed by checking for stabilization between the values obtained from $r$ to $r + 1$ as demonstrated in Table 1. We see that $d_7 = d_8 = 2 = \operatorname{corank} \mathbf{c}$ and provide the extra rows to show how the entries stabilize. In particular, this confirms that (3) is unidentifiable with 2 dimensions of unidentifiability.

We summarize this computation of the dimension of unidentifiability without first explicitly computing the input-output equations $\mathbf{c}$ in the following pseudocode.

**Method 2:** Computing dimension of unidentifiability without input-output equations
**Input:** For each $r \geq 0$, system $F_r(\mathbf{q}, \mathbf{x}, \mathbf{u}, \mathbf{y})$ consisting of the coefficients of $1$, $t$, $t^2$, ..., $t^r$ and general point $\mathbf{z}_r$ such that $F_r(\mathbf{z}_r) = 0$ where $\mathbf{q}$ consists of $m_1$ parameters.
**Output:** Dimension of unidentifibility $\ell = \operatorname{corank} \mathbf{c} = \dim \mathbf{c}^{-1}(\mathbf{c}(\mathbf{p}))$ for general $\mathbf{p} \in \mathbb{C}^{m_1}$.
For $r = 0, 1, 2, \ldots$
  Compute $d_r = \operatorname{corank}_0 JF_r(\mathbf{z}_r) - \operatorname{corank}_{m_1} JF_r(\mathbf{z}_r)$
  If either $d_r = 0$ or $r > 0$ with $\operatorname{corank}_0 JF_r(\mathbf{z}_r) = \operatorname{corank}_0 JF_{r-1}(\mathbf{z}_{r-1})$ and $\operatorname{corank}_{m_1} JF_r(\mathbf{z}_r) = \operatorname{corank}_{m_1} JF_{r-1}(\mathbf{z}_{r-1})$, return $d_r$.

Such an approach naturally extends to problems when the parameters and initial conditions are restricted to an irreducible component by simply appending to $F_r$ the requested constraints and taking the test points to be general on the corresponding irreducible component. The following demonstrates this on Example 1 from [28].

**Table 1. Summary of computations showing 2 dimensions of unidentifiability.**

| $r$ | $N_r$ | $\operatorname{corank}_0 JF_r$ | $\operatorname{corank}_7 JF_r$ | $d_r$ |
|---|---|---|---|---|
| 0 | 13 | 9 | 2 | 7 |
| 1 | 16 | 8 | 1 | 7 |
| 2 | 19 | 7 | 0 | 7 |
| 3 | 22 | 6 | 0 | 6 |
| 4 | 25 | 5 | 0 | 5 |
| 5 | 28 | 4 | 0 | 4 |
| 6 | 31 | 3 | 0 | 3 |
| 7 | 34 | 2 | 0 | 2 |
| 8 | 37 | 2 | 0 | 2 |
| 9 | 40 | 2 | 0 | 2 |
| 10 | 43 | 2 | 0 | 2 |

**Table 2. Summary of computations showing the model (5) is identifiable.**

| $r$ | $N_r$ | $\mathrm{corank}_0\, JF_r$ | $\mathrm{corank}_3\, JF_r$ | $d_r$ |
|-----|-------|---------------------------|---------------------------|-------|
| 0 | 9 | 5 | 2 | 3 |
| 1 | 12 | 4 | 1 | 3 |
| 2 | 15 | 3 | 0 | 3 |
| 3 | 18 | 2 | 0 | 2 |
| 4 | 21 | 1 | 0 | 1 |
| 5 | 24 | 0 | 0 | 0 |

**Example 7.** Consider the following three-compartment model [29]:

$$\dot{x}_1 = p_{13}x_3 + p_{12}x_2 - p_{21}x_1 + u$$

$$\dot{x}_2 = p_{21}x_1 - p_{12}x_2$$

$$\dot{x}_3 = -p_{13}x_3 \qquad\qquad (5)$$

$$y = x_2$$

with state variables $\mathbf{x}(t) = (x_1(t), x_2(t), x_3(t))$, input $u(t)$, output $y(t)$, and unknown parameters $\mathbf{p} = (p_{12}, p_{13}, p_{21})$. Using a similar setup from Ex. 6 summarized in Method 2, Table 2 shows that the model (5) is identifiable.

Let $F_r'$ be the system obtained by adding the constraint $x_3(0) = 0$ to $F_r$. Table 3 shows that the model (5) is now unidentifiable with one dimension of unidentifiability.

**Identifiable functions.** When a model (1) is unidentifiable, one can ask for functions of the parameters $\mathbf{p}$ which are actually functions of the coefficients $\mathbf{c}(\mathbf{p})$ of the input-output equations. Such functions are called *identifiable functions*. For example, every element of $\mathbf{c}$ is itself an identifiable function. This is algebraically formalized in the following.

**Definition 8.** Let $\mathbf{c}$ be as above. A real-valued function $f(\mathbf{p})$ is *identifiable* if the field extension $\mathbb{R}(f, \mathbf{c})/\mathbb{R}(\mathbf{c})$ is an algebraic field extension.

One can also consider the global and local identifiability of functions.

**Definition 9.** Let $\mathbf{c}$ be as above and $f$ be an identifiable function. The function $f$ is called *globally identifiable* from $\mathbf{c}$ if there exists a function $\phi$ such that $\phi \circ \mathbf{c} = f$. The function $f$ is called *locally identifiable* from $\mathbf{c}$ if there exists a multi-valued function $\xi$ such that for every $\mathbf{p}$, $f(\mathbf{p})$ is equal to an entry of the multi-valued function $\xi \circ \mathbf{c}(\mathbf{p})$.

**Example 10.** With the setup from Ex. 3, the function $f(\mathbf{p}) = k_{01} + k_{21}$ is globally identifiable with $f = \phi \circ \mathbf{c}$ where $\phi(x_1, \ldots, x_5) = x_4 - x_1$, i.e., $f = c_4 - c_1$. The function $g(\mathbf{p}) = k_{02} + k_{12} + k_{32}$ is

**Table 3. Summary of computations showing that (5) is unidentifiable when $x_3(0) = 0$.**

| $r$ | $N_r$ | $\mathrm{corank}_0\, JF_r'$ | $\mathrm{corank}_3\, JF_r'$ | $d_r$ |
|-----|-------|-----------------------------|-----------------------------|-------|
| 0 | 9 | 4 | 1 | 3 |
| 1 | 12 | 3 | 0 | 3 |
| 2 | 15 | 2 | 0 | 2 |
| 3 | 18 | 1 | 0 | 1 |
| 4 | 21 | 1 | 0 | 1 |

locally identifiable with $g^2 + c_4 g + c_5 = 0$, i.e., $g = \xi \circ \mathbf{c}$ where That is, for

$$\xi(x_1, \ldots, x_5) = \frac{-x_4 \pm \sqrt{x_4^2 - 4x_5}}{2},$$

we have

$$\xi \circ \mathbf{c}(\mathbf{p}) = \frac{1}{2}(k_{02} + k_{12} + k_{32} + k_{03} + k_{13} \pm |k_{02} + k_{12} + k_{32} - k_{03} - k_{13}|).$$

The entry of this 2-valued function which is equal to $g(\mathbf{p})$ is selected based on the sign of

$$k_{02} + k_{12} + k_{32} - k_{03} - k_{13},$$

i.e., the "+" entry when $k_{02} + k_{12} + k_{32} - k_{03} - k_{13} \geq 0$ and the "–" entry otherwise.

When a model is unidentifiable with $\ell = \operatorname{corank} \mathbf{c}$ dimensions of unidentifiability, the goal is to compute $d = \operatorname{rank} \mathbf{c}$ algebraically independent identifiable functions. The problem of finding $d$ "nice" algebraically independent identifiable functions can be described in the following way, where "nice" could be taken to mean low degree, sparse, or are easy to interpret in terms of the model, depending on the context.

**Problem 11**. For rational functions $\mathbf{c}$ with $d = \operatorname{rank} \mathbf{c}$, compute a "nice" transcendence basis of the field extension $\mathbb{R}(\mathbf{c})/\mathbb{R}$.

One way to locate identifiable functions is by computing Gröbner bases with respect to various elimination orderings of the ideal $\langle \mathbf{c}(\mathbf{q}) - \mathbf{c}(\mathbf{p}) \rangle$. This approach is described in [2, 30] and has been implemented in the web application COMBOS [30]. In addition to requiring $\mathbf{c}$, e.g., computed using differential elimination techniques, the biggest disadvantage of this method is that Gröbner basis computations can be computationally expensive. Thus, COMBOS can fail even for relatively simple models. Alternatively, the program DAISY [1, 24] can sometimes be used to find identifiable functions. Specifically, the DAISY program gives the solution to the system of equations $\mathbf{c}(\mathbf{q}) = \mathbf{c}(\mathbf{p})$ for a randomly chosen numerical point $\mathbf{p}$. Sometimes one can algebraically manipulate the solution to obtain functions of the form $f(\mathbf{q}) = f(\mathbf{p})$, but there are many cases where this cannot be done [2, 30]. Nonetheless, if one is able to obtain such $f$, the following shows that they are indeed identifiable functions.

**Proposition 12**. *If $f(\mathbf{q}) - f(\mathbf{p})$ is an element of the ideal $I = \langle \mathbf{c}(\mathbf{q}) - \mathbf{c}(\mathbf{p}) \rangle \subset \mathbb{R}[\mathbf{p}, \mathbf{q}]$, then $f$ is an identifiable function. That is, if $f$ is constant on irreducible components of generic fibers of $\mathbf{c}$, then $f$ is an identifiable function.*

*proof.* If $f(\mathbf{q}) - f(\mathbf{p})$ is contained in $I$, then the dimension of the image of the combined map $(\mathbf{c}, f)$ is equal to the dimension of the image of the map $\mathbf{c}$. In other words, the field extension $\mathbb{R}(f, \mathbf{c})/\mathbb{R}(\mathbf{c})$ is an algebraic field extension showing that $f$ is identifiable.

**Reparametrization and other uses of identifiable functions.** If one can solve Problem 11, one then tries to use the new basis to reparametrize the model. In [23], a method to find identifiable scaling reparametrizations is given for a certain class of linear compartment models where the identifiable functions are monomials. Currently, there is no general approach to find identifiable reparametrizations and, for most models, the reparametrizations are found using *ad hoc* approaches which work on a case-by-case basis.

Even if a reparametrization cannot be found, identifiable functions have other important uses. From the identifiable functions, one can determine which parameters need to be known in order to render the entire model identifiable. This information can also be determined from the solution of the system of equations $\mathbf{c}(\mathbf{q}) = \mathbf{c}(\mathbf{p})$. However, identifiable functions give us additional information if only a subset of those parameters can be determined. In other words, we can obtain a simpler set of identifiable functions of parameters if a subset of the parameters

are known and, perhaps, for this new set of identifiable functions, computing an identifiable reparametrization is possible. This is the case for Ex. 23 below where knowledge of either the pair $(a_{34}, a_{43})$ or the pair $(a_{33}, a_{44})$ renders all the identifiable functions to be monomials, in which case the method in [23] can be used to find an identifiable scaling reparametrization.

## Computing identifiability degree

For a model (1) that is identifiable, Problem 4 can be solved by computing the identifiability degree $k \in \mathbb{N}$ in order to distinguish between globally identifiable ($k = 1$) and locally identifiable ($k > 1$) models. $k$ is simply the number of solutions of $\mathbf{c}(\mathbf{q}) = \mathbf{c}(\mathbf{p})$ for general $\mathbf{p}$, where $\mathbf{c}$ is the collection of coefficients of the input-output equations. As mentioned above, the software package DAISY [1, 24] uses such an approach with Gröbner basis methods to count the number of solutions. One could also use numerical homotopy methods, e.g., as summarized in [6, 7], to compute $k$, as illustrated in the following example.

**Example 13** As shown in Ex. 3, the model (3) has 2 dimensions of unidentifiability. With the aim of constructing an identifiable model, we modify (3) by adding the extra constraints $k_{01} = k_{03} = 0$ yielding a new model with only one leak parameter $k_{02}$. The coefficients $\mathbf{c}$ of the input-output equation for this simplified model are

$$
\mathbf{c}(k_{02}, k_{12}, k_{13}, k_{21}, k_{32}) = \begin{bmatrix} k_{02} + k_{12} + k_{13} + k_{21} + k_{32} \\ k_{02}k_{13} + k_{02}k_{21} + k_{12}k_{13} + k_{13}k_{21} + k_{13}k_{32} + k_{21}k_{32} \\ k_{02}k_{13}k_{21} \\ k_{02} + k_{12} + k_{13} + k_{32} \\ k_{02}k_{13} + k_{12}k_{13} + k_{13}k_{32} \end{bmatrix}, \tag{6}
$$

which is easily seen to have rank 5, i.e., the model is identifiable. For general $\alpha_i \in \mathbb{C}$, the system

$$
\mathbf{c}(k_{02}, k_{12}, k_{13}, k_{21}, k_{32}) - \mathbf{c}(\alpha_1, \alpha_2, \alpha_3, \alpha_4, \alpha_5) = 0 \tag{7}
$$

consists of 5 equations (1 cubic, 2 quadratic, and 2 linear) in 5 variables. Using a total degree homotopy (see [6, 7] for more details), one tracks $3 \cdot 2^2 \cdot 1^2 = 12$, i.e., the total degree of (7), solution paths. Tracking these paths with homotopy continuation, e.g., via Bertini [31], yields 2 solutions to (7). One can also use a Gröbner basis computation to see that (7) has 2 solutions. These computations show that the model (3) with $k_{01} = k_{03} = 0$ is locally identifiable with identifiability degree of 2.

We summarize this most basic approach for computing the identifiability degree when the input-output equations are known in the following pseudocode.

**Method 3:** Computing identifiability degree from input-output equations (direct solving)
**Input:** $m_2$ input-output equation coefficients $\mathbf{c}(\mathbf{q})$, depending on parameters $\mathbf{q} \in \mathbb{C}^{m_1}$ for which corank $\mathbf{c} = 0$, i.e., corresponding model is identifiable.
**Output:** Identifiability degree $k \in \mathbb{N}$.
Choose random, complex values $\mathbf{p}$ of parameters $\mathbf{q}$.
Use homotopy continuation to compute $Z = \{\mathbf{q} \in \mathbb{C}^{m_1} \mid \mathbf{c}(\mathbf{q}) = \mathbf{c}(\mathbf{p})\}$.
Return $k = \#Z$.

Rather than using a direct global solving method which is based on knowing the coefficients $\mathbf{c}$, we next consider an alternative approach based on monodromy computations in numerical algebraic geometry that also can be used without computing $\mathbf{c}$. We first describe the approach when $\mathbf{c}$ is known and then extend to the case when $\mathbf{c}$ is not explicitly computed.

**Identifiability degree from input-output equations.**   Suppose that (1) is identifiable with identifiability degree $k \in \mathbb{N}$ and $\mathbf{c}$ is the set of coefficients of the input-output equations. Following the notation before Definition 1, let $m_1$ be the number of independent parameters $\mathbf{p}$ and $m_2$ be the number of entries in $\mathbf{c}$ so that $\mathbf{c} : \mathbb{C}^{m_1} \to \mathbb{C}^{m_2}$. Assume that the model is identifiable so that corank $\mathbf{c} = 0$ and rank $\mathbf{c} = \dim X = m_1$ where $X = \overline{\mathbf{c}(\mathbb{C}^{m_1})} \subset \mathbb{C}^{m_2}$. The continuity of $\mathbf{c}$ yields that $X$ is irreducible. The graph of $\mathbf{c}$, namely

$$\text{Graph}(\mathbf{c}) = \{(\mathbf{p}, \mathbf{c}(\mathbf{p})) \mid \mathbf{p} \in \mathbb{C}^{m_1}\} \subset \mathbb{C}^{m_1} \times \mathbb{C}^{m_2}$$

is also irreducible of dimension $m_1$. In fact, for the projection map $\pi : \mathbb{C}^{m_1} \times \mathbb{C}^{m_2} \to \mathbb{C}^{m_2}$, we know that $X = \overline{\pi(\text{Graph}(\mathbf{c}))}$ and $\pi$ restricted to $\text{Graph}(\mathbf{c})$ is generically a $k$-to-1 map.

One can compute $k$ via a pseudowitness point set [27] for $X$. To that end, let $\mathcal{L}_2 \subset \mathbb{C}^{m_2}$ be a general linear space of codimension $m_1$. The finite set $W = \text{Graph}(\mathbf{c}) \cap (\mathbb{C}^{m_1} \times \mathcal{L}_2)$ is a *pseudowitness point set* for $X$ with respect to $\text{Graph}(\mathbf{c})$ and $\pi$ where $\#W = k \cdot \deg X$ and $\#\pi(W) = \deg X$, i.e., $k = \#W / \#\pi(W)$. In order to compute $W$, we follow the approach in [32] using monodromy loops [33], as follows.

We first note that it is trivial to construct one point $w \in W$ as follows. One first selects a general point $(\mathbf{p}, \mathbf{c}(\mathbf{p})) \in \text{Graph}(\mathbf{c})$ and then constructs a general linear space $\mathcal{L}_2 \subset \mathbb{C}^{m_2}$ of codimension $m_1$ that passes through $\mathbf{c}(\mathbf{p})$. Hence, $w = (\mathbf{p}, \mathbf{c}(\mathbf{p})) \in W$.

Next, the irreducibility of $\text{Graph}(\mathbf{c})$ ensures that pairs of points in $W$ are connected via smooth paths on $\text{Graph}(\mathbf{c})$. We aim to discover such connecting paths using random monodromy loops. For $t \in [0, 1]$, let $\mathcal{L}(t)$ be a smooth path consisting of general linear spaces of codimension $m_1$ in $\mathbb{C}^{m_2}$ such that $\mathcal{L}(0) = \mathcal{L}(1) = \mathcal{L}_2$. Hence, this defines paths $w(t)$ defined by $\text{Graph}(\mathbf{c}) \cap (\mathbb{C}^{m_1} \times \mathcal{L}(t))$ where $w(1) \in W$ is known. Homotopy continuation computes the endpoint $w(0)$, which must also be a point in $W$. If $w(0) \neq w(1)$, the resulting loop has produced a nontrivial monodromy action and potentially yielded a previously unknown point in $W$.

**Example 14**. For $\mathbf{c} : \mathbb{C}^5 \to \mathbb{C}^5$ in (6), we know that $X = \overline{\mathbf{c}(\mathbb{C}^5)} = \mathbb{C}^5$, i.e., $\deg X = 1$. Hence, we have that the identifiability degree $k = \#W$ where $W$ is a pseudowitness point set for $X$.

For illustrative purposes, consider $\mathbf{p} = (-1, -2, 5, -1, -3)$ with $\mathbf{c}(\mathbf{p}) = (-2, -31, 5, -1, -30)$ so $\mathcal{L}_2 = \{(-2, -31, 5, -1, -30)\}$ has codimension 5 with $\mathbf{c}(\mathbf{p}) \in \mathcal{L}_2$. Consider the loop

$$\mathcal{L}(t) = \{(-2, -31 - 15s(t), 5 + 5s(t), -1, -30 + 35s(t))\}$$

where $s(t) = 1 - e^{2\pi i(1-t)}$ and $i = \sqrt{-1}$. Hence, $\mathcal{L}(t)$ is a loop with $\mathcal{L}(0) = \mathcal{L}(1) = \mathcal{L}_2$. For the path $w(t) \in \text{Graph}(\mathbf{c}) \cap (\mathbb{C}^5 \times \mathcal{L}(t))$ with $w(1) = (\mathbf{p}, \mathbf{c}(\mathbf{p}))$, we have $w(0) = (\mathbf{q}, \mathbf{c}(\mathbf{q}))$ where $\mathbf{q} = (5/6, -2, -6, -1, 37/6)$ and $\mathbf{c}(\mathbf{q}) = \mathbf{c}(\mathbf{p})$ showing $\{w(0), w(1)\} \subset W$ and $k = \#W \geq 2$.

Running finitely many random monodromy loops necessarily yields a set $\widehat{W} \subset W$ that may fail to achieve the goal of equality. However, information about the model can be obtained even if $\widehat{W} \subsetneq W$. For example, if $(\mathbf{p}_1, \mathbf{c}(\mathbf{p}_1))$ and $(\mathbf{p}_2, \mathbf{c}(\mathbf{p}_2))$ are known points in $W$ with $\mathbf{c}(\mathbf{p}_1) = \mathbf{c}(\mathbf{p}_2)$ and $\mathbf{p}_1 \neq \mathbf{p}_2$, then one knows the identifiability degree is larger than 1, i.e., the model is locally identifiable. A heuristic stopping criterion for when $\widehat{W} = W$ provided in [32] is simply to have many different random monodromy loops yielding no new points.

We use trace tests [34, 35] to provide a stopping criterion to recognize when $\widehat{W} = W$. These are described and illustrated well in [36]. To make these monodromy and trace test computations more efficient, see [37, 38].

**Example 15**. To show that Ex. 14 computed both points in $W$, i.e., the degree of identifiability $k = 2$, for illustrative purposes, we consider the following three linear spaces in $\mathbb{C}^5$:

$$\mathcal{H}_1 = \{4x_1 + 5x_2 - 2x_3 + 4x_4 - 2x_5 - 1 = 0\},$$

$$\mathcal{H}_2 = \{2y_1 + 4y_2 - y_3 - 6y_4 - 4y_5 + 7 = 0\}, \quad \text{and}$$

$$\mathcal{M}_2 = \{(-2 - 5r, -31 - 3r, 5 - 3r, -1 + 2r, -30 + 4r) \mid r \in \mathbb{C}\},$$

with $\mathcal{L}_2 = \mathcal{M}_2 \cap \mathcal{H}_2$. We take

$$\mathcal{H}(t) = \{(4x_1 + 5x_2 - 2x_3 + 4x_4 - 2x_5 - 1)(2y_1 + 4y_2 - y_3 - 6y_4 - 4y_5 + 7) - 2t = 0\} \subset \mathbb{C}^5 \times \mathbb{C}^5.$$

The irreducible curve $\mathcal{C} = \mathrm{Graph}(\mathbf{c}) \cap (\mathbb{C}^5 \times \mathcal{M}_2)$ has multidegree $(5, 2)$, which is verified using the multihomogeneous trace test applied to $\mathcal{C} \cap \mathcal{H}(t)$. This yields $k = 2$.

We summarize this computation in the following pseudocode.

**Method 4:** Computing identifiability degree from input-output equations (monodromy)

```
Input: m₂ input-output equation coefficients c(q), depending on param-
eters q ∈ ℂᵐ¹ for which corank c = 0, i.e., model is identifiable, and
an integer maxUselessLoops.
Output: Identifiability degree k ∈ ℕ or error along with a lower bound
on k if the number of loops in a row that do not yield any new points
is more than maxUselessLoops.
Choose random, complex values p of parameters q and compute c(p).
Form w = (p, c(p)) and W = {w}.
Construct general linear space L₂ ⊂ ℂᵐ² of codimension m₁ that passes
through c(p).
Set numUselessLoops = 0.
While numUselessLoops < maxUselessLoops
  Increment numUselessLoops = numUselessLoops + 1.
  Construct a general loop of linear spaces L(t) such that
L(0) = L(1) = L₂.
  For each w ∈ W
    Use homotopy continuation applied to the homotopy Graph(c) ∩ (ℂᵐ¹ ×
L(t)) to
    track from w at t = 1 to t = 0 yielding w′.
  If w′ ∉ W
    Update w = {W, w′} and numUselessLoops = 0.
    If trace test passes, return k = #W.
Return error with k = #W.
```

The advantage to using such a monodromy approach is that the structure of $\mathbf{c}$ may be such that $k$ is small but this structure is not known *a priori* meaning that a homotopy for solving $\mathbf{c}(\mathbf{q}) = \mathbf{c}(\mathbf{p})$ requires tracking many homotopy paths. The disadvantage is that many monodromy loops may be needed to find all points necessary for the trace test to validate completeness when $k$ is large.

**Identifiability degree without input-output equations.** In the previous section, we computed the degree of a general fiber of a generically finite-to-one coefficient map. This is based on the fact that one has the same input-output equation if and only if the coefficients agree. However, when we are using a truncated system as described in Example 6, namely $F_r$ which depends upon the parameters $\mathbf{p}$, input $\mathbf{U} = \{\mathbf{u}_0, \ldots, \mathbf{u}_r\}$, output $\mathbf{Y} = \{\mathbf{y}_0, \ldots, \mathbf{y}_r\}$, and state variables $\mathbf{X} = \{\mathbf{x}_0, \ldots, \mathbf{x}_{r+1}\}$, it provides necessary conditions to have the same input-output as shown in the following example.

**Table 4. Summary of computations showing (8) is identifiable.**

| $r$ | $\mathrm{corank}_0 \, JF_r$ | $\mathrm{corank}_0 \, JF_r$ | $d_r$ |
|---|---|---|---|
| 0 | 7 | 2 | 5 |
| 1 | 6 | 1 | 5 |
| 2 | 5 | 0 | 5 |
| 3 | 4 | 0 | 4 |
| 4 | 3 | 0 | 3 |
| 5 | 2 | 0 | 2 |
| 6 | 1 | 0 | 1 |
| 7 | 0 | 0 | 0 |

**Example 16**. The following model is a modification of an HIV model from [39]:

$$\dot{x}_1 = p_1 - p_2 x_1 - p_3 x_1 x_3$$

$$\dot{x}_2 = p_3 x_1 x_3 - p_4 x_2$$

$$\dot{x}_3 = p_1 p_4 x_2 - p_5 x_3 \tag{8}$$

$$y = x_3$$

As in the previous section, Table 4 shows that the system $F_7$ provides the model (8) is identifiable.

For example, consider the sufficiently general truncated output

$$\mathbf{Y} = (y_0, \ldots, y_7) = (0.5, -0.03, -0.15, -0.2, -0.2, -0.17, -0.16, -0.15).$$

We know that there are finitely many values of the parameters $\mathbf{p}$ which yield this output. Monodromy yields the following 12 values of the parameters (listed to four decimal places):

This table shows that there are 3 distinct values of $y_8$, each of which is obtained by 4 values of the parameters indicating that the identifiability degree is 4.

This example shows that even though $F_r$ is enough to show identifiability, we may only need to consider a subset of the corresponding parameter values which have the same input-output.

The structure of (1) clearly shows that the solution set of $F_r = 0$ is irreducible, smooth, and parameterized by $\mathbf{p}$, $\mathbf{U}$, and $\mathbf{x}_0$. Thus, it is trivial to construct a generic point $(\mathbf{p}^*, \mathbf{X}^*, \mathbf{U}^*, \mathbf{Y}^*)$ in the solution set of $F_r = 0$. From this point, we can use Prop. 5 to compute the dimension $d \geq 0$ of the solution set of $F_r(\mathbf{p}^*, \mathbf{X}, \mathbf{U}^*, \mathbf{Y}^*) = 0$, i.e., the dimension of the state variables. If $d > 0$, we can add $d$ general linear slices in $\mathbf{X}$ to $F_r$ to reduce to the case when $d = 0$.

With this reduction, we repeatedly apply random monodromy loops to compute all values of $\mathbf{p}$ such that there exists $\mathbf{X}$ with

$$F_r(\mathbf{p}, \mathbf{X}, \mathbf{U}^*, \mathbf{Y}^*) = 0.$$

By testing the finitely many values of the parameters $\mathbf{p}$, the identifiability degree $k$ is the number of points corresponding to the same input-output. To verify the completeness, we simply apply the multihomogeneous trace test via the parameter space and the input-output space.

To save space, we exclude pseudocode for this method as it is so similar to Method 1. The primary change is that the set of coefficients $\mathbf{c}$ is replaced by the truncated system $F_r$ for some value of $r$ along with an extra computation to test for the same input-output values.

**Table 5. 12 possible values of parameters of system $F_7$ from model (8).**

| $p_1$ | $p_2$ | $p_3$ | $p_4$ | $p_5$ | $y_8$ |
|---|---|---|---|---|---|
| $\pm 0.1253$ | $-2.4825$ | $4.4249$ | $-0.9210$ | $-0.2137$ | $0.1706$ |
| $\pm 0.2602$ | $-2.4825$ | $4.4249$ | $-0.2137$ | $-0.9210$ | $0.1706$ |
| $0.3023 \pm 0.0779i$ | $-3.5234 \pm 0.5105i$ | $4.2201 \pm 1.9168i$ | $-1.3367 \mp 0.0298i$ | $-0.1080 \mp 0.2292i$ | $0.1107 \mp 0.4040i$ |
| $-0.3023 \pm 0.0779i$ | $-3.5234 \mp 0.5105i$ | $4.2201 \mp 1.9168i$ | $-1.3367 \pm 0.0298i$ | $-0.1080 \pm 0.2292i$ | $0.1107 \pm 0.4040i$ |
| $0.6847 \pm 0.2133i$ | $-3.5234 \mp 0.5105i$ | $4.2201 \mp 1.9168i$ | $-0.1080 \pm 0.2292i$ | $-1.3367 \pm 0.0298i$ | $0.1107 \pm 0.4040i$ |
| $-0.6847 \pm 0.2133i$ | $-3.5234 \pm 0.5105i$ | $4.2201 \pm 1.9168i$ | $-0.1080 \mp 0.2292i$ | $-1.3367 \mp 0.0298i$ | $0.1107 \mp 0.4040i$ |

**Example 17**. Reconsidering the model (8) in Ex. 16 which has no input, we first restrict the output space to, for illustration purposes, the sufficiently general line

$$\mathbf{Y}(s) = (s + 0.5, 4s - 0.03, 3s - 0.15, -2s - 0.2, -s - 0.2, -3s - 0.17, 3s - 0.16, 4s - 0.15).$$

Thus, we apply the multihomogeneous trace test by solving $F_7 = 0$ on this line intersected with the sufficiently general family of bilinear hyperplanes in the parameter and output space:

$$\mathcal{H}(t) = \{(2p_1 - 3p_2 - 4p_3 - p_4 - 4p_5 - 5)(3y_0 + 4y_1 + 5y_2 + y_3 + y_4 - 4y_5 + 4y_6 - y_7 - 0.42) - t = 0\}.$$

Monodromy followed by the trace test confirms that the bidegree is (222, 12). Hence, the number of elements in Table 5 is complete.

We can simplify this computation, for example, by instead taking the following family

$$\mathcal{H}(t) = \{(3p_5 - 4)(3y_0 + 4y_1 + 5y_2 + y_3 + y_4 - 4y_5 + 4y_6 - y_7 - 0.42) - t = 0\}.$$

The bidegree in $p_5$ and the output space is (60, 12) which again shows that Table 5 is complete.

## Computing identifiable functions

A model (1) is identifiable if and only if every function of the parameters is an identifiable function. In particular, each irreducible component of a generic fiber of the coefficients $\mathbf{c}$ of the input-output equations is a singleton for an identifiable model. Since every function of the parameters is trivially constant on each singleton, Prop. 2 yields that every function is identifiable. To be a globally identifiable function, it must take the same constant value on all of the irreducible components of a general fiber.

**Example 18**. With the setup from Ex. 16, the model (8) is identifiable with identifiability degree 4. Hence, for example, we know that $f_1 = p_4$ and $f_2 = p_5$ are both identifiable functions. From the first two rows of Table 5, we see that both $f_1$ and $f_2$ are not globally identifiable since each of them take two different values. The functions $g_1 = p_2$, $g_2 = p_3$, and $g_3 = p_4 + p_5$ are all globally identifiable since each of them take the same value at all four points.

To compute identifiable functions, we will first use numerical algebraic geometry to sample points from fibers. Then, given a finite collection of terms, we will use exactness recovery techniques, e.g., [40], or interpolation to construct identifiable functions from the sample data. Computing globally identifiable functions simply requires computing points on all irreducible components and adding additional constraints.

**Sampling.** In the case that input-output equations have been computed, let $\mathbf{c}$ be the collection of coefficients of the input-output equations and suppose that $d \geq 0$ is the dimension of unidentifiability. Thus, for a given generic point $\mathbf{p}$, the point $\mathbf{q} = \mathbf{p}$ is a smooth point on an irreducible component $V_{\mathbf{p}}$ of dimension $d$ of the solution set defined by $\mathbf{c}(\mathbf{q}) - \mathbf{c}(\mathbf{p}) = 0$. Hence, when $d > 0$, we can sample other points in this irreducible component as follows. Let $\mathcal{L}_{\mathbf{p}}$ be a

general linear space of codimension $d$ passing through $\mathbf{p}$ and $\mathcal{L}$ be some other general linear space of codimension $d$. By using homotopy continuation, we can track the solution path $\mathbf{q}(t)$ defined by $\mathbf{q}(1) = \mathbf{p}$ and

$$\mathbf{c}(\mathbf{q}(t)) - \mathbf{c}(\mathbf{p}) = 0$$

$$\mathbf{q}(t) \in t \cdot \mathcal{L}_{\mathbf{p}} + (1 - t) \cdot \mathcal{L}.$$

(9)

This yields the point $\mathbf{q}(0)$ which is also a generic point in $V_{\mathbf{p}}$.

One can easily compute other points in this same fiber $V_{\mathbf{p}}$ by repeating with a different linear space $\mathcal{L}$ and sample other fibers by repeating the process with a different generic point $\mathbf{p}$.

With the aim of computing globally identifiable functions, sample points in every irreducible component of $\mathbf{c}(\mathbf{q}) - \mathbf{c}(\mathbf{p}) = 0$ are needed. In this case, one simply constructs an identifiable system by restricting the parameters to a general linear space of codimension $d$ and applying the techniques above to the resulting system. That is, if $\mathbf{p} \in \mathbb{C}^{m_1}$, we take a general affine linear mapping $\mathbf{b} : \mathbb{C}^{m_1 - d} \to \mathbb{C}^{m_1}$ so that $\mathbf{c}(\mathbf{b}(\hat{\mathbf{q}})) - \mathbf{c}(\mathbf{b}(\hat{\mathbf{p}})) = 0$ has finitely many solutions for generic $\hat{\mathbf{p}}$, say $\mathbf{q}_1 = \mathbf{b}(\hat{\mathbf{q}}_1), \ldots, \mathbf{q}_k = \mathbf{b}(\hat{\mathbf{q}}_k)$, i.e., the model with parameters $\mathbf{p} = \mathbf{b}(\hat{\mathbf{p}})$ is identifiable over $\mathbb{C}^{m_1 - d}$ with identifiability degree $k$. Applying the slice moving described above, one can sample points in all components of the fiber over $\mathbf{p}$ using the points $\mathbf{q}_1, \ldots, \mathbf{q}_k$.

**Example 19**. Reconsider (3) in Ex. 3 for which $\mathbf{c}$ shows the model has $d = 2$ dimensions of unidentifiability. For illustration, with $\mathbf{p} = (1, 2, 3, 4, 5, 6, 7)$, we can take $\mathcal{L}_{\mathbf{p}}$ to be

$$\{k_{01} - k_{02} + k_{03} - k_{12} + k_{13} - k_{21} + k_{32} = 4, 2k_{02} - k_{01} + 2k_{03} + k_{12} - k_{13} - 2k_{21} + 2k_{32} = 10\} \quad (10)$$

and

$$\mathcal{L} = \left\{ \begin{array}{r} k_{01} + k_{02}(3 - i) + k_{03}(-3 + 2i) + k_{12}(1 + i) - k_{13}(2 + 2i) + k_{21}(2 - i) - 2k_{32} = 1 \\ k_{01}(1 - 3i) - 3ik_{02} + k_{03}(-2 + 2i) - 2k_{12} - ik_{13} + k_{21}(3 + 2i) - ik_{32} = 1 \end{array} \right\} \quad (11)$$

where $i = \sqrt{-1}$. Tracking the path defined by (9) yields the endpoint (to four decimal places):

$$(0.6709 - 2.1940i, \; 3.6921 + 2.5919i, \; 2.8774 + 0.5068i, \; 3.3852 - 1.1735i,$$

$$5.1226 - 0.5068i, \; 6.3291 + 2.1940i, \; 5.9227 - 1.4185i). \quad (12)$$

Hence, since all of the values of the parameters changed, we know that each parameter itself is an unindentifiable function.

If, for illustration, we take the affine linear mapping $\mathbf{b} : \mathbb{C}^5 \to \mathbb{C}^7$ defined by

$$\mathbf{b}(\hat{\mathbf{p}}) = \begin{bmatrix} \hat{\mathbf{p}}_1 \\ \hat{\mathbf{p}}_2 \\ \hat{\mathbf{p}}_3 \\ \hat{\mathbf{p}}_4 \\ \hat{\mathbf{p}}_5 \\ \hat{\mathbf{p}}_1 + 3\hat{\mathbf{p}}_2 - \hat{\mathbf{p}}_3 - 3\hat{\mathbf{p}}_4 + 2\hat{\mathbf{p}}_5 + 4 \\ 2\hat{\mathbf{p}}_1 + 3\hat{\mathbf{p}}_2 + 5\hat{\mathbf{p}}_3 + \hat{\mathbf{p}}_4 - 3\hat{\mathbf{p}}_5 - 5 \end{bmatrix}$$

the resulting model is identifiable with identifiability degree 8 and the following 7 other points

corresponding with $\mathbf{b}(1, 2, 3, 4, 5) = (1, 2, 3, 4, 5, 6, 7)$:

$$(9.2814, -10.3208, 10.7201, -10.52, -2.7201, -2.2814, 33.8409),$$

$$(108.0762, -66.9431, 13.0118, -0.23744, -0.0118, -101.0762, 75.1805),$$

$$(2.4938, 0.3612, 4.3645, 5.326, 8.6355, 4.5062, 2.3128),$$

$$(52.0709, -31.4763, 8.1035, -0.5325, -0.1035, -45.0709, 45.0087), \tag{13}$$

$$(8.6814, -13.22, 14.5081, -14.2737, -1.5081, -1.6814, 35.4937),$$

$$(-9.615, 5.8203, 5.6325, 1.4445, 7.3675, 16.615, 0.7352),$$

$$(-13.625, 9.5193, 3.6057, 1.1636, 4.3943, 20.625, 2.317).$$

Thus, we have computed at least one point in each irreducible component of the fiber over $\mathbf{p}$.

Without input-output equations, one simply uses a truncated system $F_r$ as described in Example 6 to perform the same computations. The only potential issues were addressed above, namely reduction to the case that the state variables are generically zero-dimensional over the parameter-input-output space and restricting to the irreducible components which have the same input-output. The latter is accomplished by simply ignoring the components which have different input-output values.

**Example 20**. To illustrate moving on an irreducible component, we describe the setup to yield the same corresponding endpoint in (12). To that end, following Ex. 6, we utilize $F_7$. Starting with parameter values $\mathbf{p} = (1, 2, 3, 4, 5, 6, 7)$, the structure of $F_7$ makes it trivial to generate general input, output, and state variables satisfying $F_7 = 0$, i.e., randomly selecting input $\mathbf{U}$ and initial conditions $\mathbf{x}_0$ for the state variables trivially yields the values of the other state variables $\mathbf{x}_1, \ldots, \mathbf{x}_8$ and output $\mathbf{Y}$. Then, by holding the input $\mathbf{U}$ and output $\mathbf{Y}$ fixed, we track along the solution path where the variables consist of the model parameters and the state variables defined by $F_7 = 0$ that deforms $\mathcal{L}_{\mathbf{p}}$ in (10) to $\mathcal{L}$ in (11). The resulting endpoint corresponds with the endpoint in (12).

**Functions from samples.** From the ability to sample points described in the previous section, we can reconstruct identifiable functions in a given finite-dimensional vector space of functions, say $\mathcal{F} = \text{span}\{f_1, \ldots, f_j\}$. Following Prop. 2, an identifiable function $f \in \mathcal{F}$ is constant on irreducible components of generic fibers of $\mathbf{c}$, which corresponds with computing null spaces of linear equations described as follows.

We can express every $f \in \mathcal{F}$ as $f = \sum_{i=1}^{j} a_i f_i$ where $\mathbf{a} = (a_1, \ldots, a_j) \in \mathbb{C}^j$. If $\mathbf{p}$ is a generic value of the parameters, using the sampling method above, we can compute a generic $\mathbf{q}_{\mathbf{p}}$ in the same irreducible component $V_{\mathbf{p}}$. Hence, the condition $f(\mathbf{q}_{\mathbf{p}}) = f(\mathbf{p})$ imposes a linear constraint on $\mathbf{a}$, namely

$$\left[ f_1(\mathbf{q}_{\mathbf{p}}) - f_1(\mathbf{p}) \quad \cdots \quad f_j(\mathbf{q}_{\mathbf{p}}) - f_j(\mathbf{p}) \right] \cdot \mathbf{a} = 0.$$

One option is to keep imposing more such conditions by selecting other general values of $\mathbf{p}$ with corresponding $\mathbf{q}_{\mathbf{p}}$. The dimension of the null space reduces by one monotonically with each new condition until it reaches the dimension of the linear span of the identifiable functions in $\mathcal{F}$.

Alternatively, for computing identifiable functions with integer coefficients, i.e., $\mathbf{a} \in \mathbb{Z}^j$, one general point is enough via exactness recovery methods [40].

**Example 21**. Let $\mathcal{F} = \mathrm{span}\{k_{01}, k_{02}, k_{03}, k_{12}, k_{13}, k_{21}, k_{32}\}$, $\mathbf{p} = (1, 2, 3, 4, 5, 6, 7)$, and $\mathbf{q_p}$ as in (12). Then, integer solutions to $(\mathbf{q_p} - \mathbf{p}) \cdot \mathbf{a} = 0$ computed using [40] correspond to:

$$k_{01} + k_{21}, \quad k_{03} + k_{13}, \quad k_{02} + k_{12} + k_{32}.$$

Alternatively, one can sample $V_\mathbf{p}$ for five general values of $\mathbf{p}$ and observe that the first four impose a new linear constraint on the coefficients $\mathbf{a}$ while the fifth one is redundant. This shows that there is a three-dimensional linear space of identifiable functions in $\mathcal{F}$ spanned by the three linear functions above.

We bring all methods of this section together in the following brief high-level pseudocode.

**Method 5:** Computing identifiable functions via sampling

```
Input: Input-output equation coefficients c(q), depending on parame-
ters q ∈ ℂ^m₁ (if available), else the truncated system F_r for some r and
a basis f₁, ..., f_j for a linear space of polynomials F of interest.
Output: Identifiable functions in F.
Choose random, complex values p of parameters q.
Compute a point on each irreducible component of c⁻¹(c(p)) using either
c or F_r.
Use homotopy sampling to find additional points on each irreducible
component.
Use the sample points together with exactness recovery methods to find
all identifiable
 functions in F.
Return all discovered identifiable functions.
```

Globally identifiable functions are computed by simply adding the condition that the function takes the same constant value on *all* irreducible components of general fibers which are sampled using the methods described above. Since globally identifiable functions are a subset of the identifiable functions, one need only search inside of the space of identifiable functions in $\mathcal{F}$.

**Example 22**. From the seven points in (13) corresponding with $\mathbf{p} = (1, 2, 3, 4, 5, 6, 7)$, we see that $k_{01} + k_{21}$ is globally identifiable (always taking the value 7 on these eight points) whereas $k_{03} + k_{13}$ and $k_{02} + k_{12} + k_{32}$ are not globally identifiable. However, from the sample points, it is easy to see that their sum, namely $k_{02} + k_{03} + k_{12} + k_{13} + k_{32}$, is globally identifiable.

The selection of the test space $\mathcal{F}$ is a user-defined input and is based on the structure of the identifiable functions of interest, e.g., linear functions, polynomials of low degree, or linear span of rational monomials where the numerator and denominator have low degree.

## Results

We now demonstrate our methods on two larger examples. Throughout the paper, for illustrative purposes, the examples presented typically selected small integer values for random numbers. In practice, including the following examples, we select random complex numbers. Data for computations available at http://dx.doi.org/10.7274/R03T9F91.

**Example 23**. The following is a 4-compartment model from Example 6.3 of [23]:

$$
\begin{aligned}
\dot{x}_1 &= a_{11}x_1 + a_{12}x_2 + u \\
\dot{x}_2 &= a_{21}x_1 + a_{22}x_2 + a_{23}x_3 \\
\dot{x}_3 &= a_{33}x_3 + a_{34}x_4 \\
\dot{x}_4 &= a_{42}x_2 + a_{43}x_3 + a_{44}x_4 \\
y &= x_1.
\end{aligned}
$$

This model, which has parameters $\mathbf{p} = (a_{11}, a_{12}, a_{21}, a_{22}, a_{23}, a_{33}, a_{34}, a_{42}, a_{43}, a_{44})$, input $u(t)$, state variables $x_1(t), x_2(t), x_3(t), x_4(t)$, and output $y(t)$, does not fit the criteria presented

in [23] for computing identifiable functions. Nonetheless, the method provided in [17, 23] is able to compute the input-output equations where the set $\mathbf{c} : \mathbb{C}^{10} \to \mathbb{C}^7$ of coefficients is

$$a_{11}a_{23}a_{34}a_{42} + a_{12}a_{21}a_{34}a_{43} - a_{11}a_{22}a_{34}a_{43} - a_{12}a_{21}a_{33}a_{44} + a_{11}a_{22}a_{33}a_{44}$$

$$a_{12}a_{21}a_{33} - a_{11}a_{22}a_{33} - a_{23}a_{34}a_{42} + a_{11}a_{34}a_{43} + a_{22}a_{34}a_{43} + a_{12}a_{21}a_{44} - a_{11}a_{22}a_{44} - \ldots$$

$$\ldots a_{11}a_{33}a_{44} - a_{22}a_{33}a_{44}$$

$$-a_{12}a_{21} + a_{11}a_{22} + a_{11}a_{33} + a_{22}a_{33} - a_{34}a_{43} + a_{11}a_{44} + a_{22}a_{44} + a_{33}a_{44}$$

$$-a_{11} - a_{22} - a_{33} - a_{44}$$

$$a_{23}a_{34}a_{42} - a_{22}a_{34}a_{43} + a_{22}a_{33}a_{44}$$

$$-a_{22}a_{33} + a_{34}a_{43} - a_{22}a_{44} - a_{33}a_{44}$$

$$a_{22} + a_{33} + a_{44}.$$

Using Prop 2, the model is unidentifiable with 4 dimensions of unidentifiability. Therefore, to solve Problem 11, we need to compute 6 algebraically independent identifiable functions.

We utilize the sampling and interpolation methods above to sample and construct the identifiable functions. For example, sampling yields two values of the parameters, provided in Table 6 rounded to four decimal places, so that every identifiable function must take the same value on both. In particular, we immediately see that both $f_1 = a_{11}$ and $f_2 = a_{22}$ are identifiable. Applying interpolation as above to the space of linear forms also yields the identifiable linear function $f_3 = a_{33} + a_{44}$.

Considering the space of polynomials of degree at most 2 which are algebraically independent of $f_1, f_2, f_3$ yields $f_4 = a_{12}a_{21}$ and $f_5 = a_{33}a_{44} - a_{34}a_{43}$.

Finally, the space of polynomials of degree at most 3 which are algebraically independent of $f_1, \ldots, f_5$ yields $f_6 = a_{23}a_{34}a_{42}$.

To show that $f_1, \ldots, f_6$ are actually globally identifiable, we use the approach above to sample points from every irreducible component. The result of this process is that a generic fiber only has one irreducible component thereby showing global identifiability. We could also have

**Table 6. Two values of the parameters rounded to four decimal places.**

| | | |
|---|---|---|
| $a_{11}$ | $-0.6690 - 0.1758i$ | $-0.6690 - 0.1758i$ |
| $a_{12}$ | $-0.1669 + 0.3165i$ | $1.3705 - 0.4117i$ |
| $a_{21}$ | $2.3433 + 0.6225i$ | $-0.5219 + 0.3086i$ |
| $a_{22}$ | $-0.6286 - 0.1868i$ | $-0.6286 - 0.1868i$ |
| $a_{23}$ | $0.4005 - 0.5144i$ | $2.5585 + 0.5746i$ |
| $a_{33}$ | $2.1248 - 0.6011i$ | $0.2095 - 0.4521i$ |
| $a_{34}$ | $1.1295 - 0.8604i$ | $0.8611 + 0.5272i$ |
| $a_{42}$ | $-0.4210 + 0.6785i$ | $0.2734 - 0.0567i$ |
| $a_{43}$ | $-1.1126 - 0.0416i$ | $-0.1132 - 0.7724i$ |
| $a_{44}$ | $-0.6880 + 0.3317i$ | $1.2273 + 0.1827i$ |

used Defn. 9 to show global identifiability. This is demonstrated by the following:

$$f_1 = -(c_4 + c_7)$$

$$f_2 = \frac{c_6^2 + c_6 c_7^2 + c_4 c_6 c_7 + c_3 c_6 + c_5 c_7 + c_1 + c_4 c_5}{c_2 + c_5 + c_4 c_6 + c_6 c_7}$$

$$f_3 = -\frac{c_2 + c_5 + c_4 c_6 + c_6 c_7}{c_7^2 + c_4 c_7 + c_3 + c_6}$$

$$f_4 = -(c_7^2 + c_4 c_7 + c_3 + c_6)$$

$$f_5 = \frac{c_1 + c_4 c_5 + c_5 c_7}{c_7^2 + c_4 c_7 + c_3 + c_6}$$

$$f_6 = -\frac{\begin{aligned}&c_1^2 + 2c_1 c_4 c_5 + c_1 c_4 c_6 c_7 + 2c_1 c_5 c_7 + c_1 c_6^2 + c_1 c_6 c_7^2 + c_3 c_1 c_6 + c_4^2 c_5^2 \\ &+ \; c_4 c_5^2 c_7 - c_2 c_4 c_5 c_7 - c_5^2 c_6 - c_3 c_5^2 - c_2 c_5 c_6 - c_2 c_5 c_7^2 - c_2 c_3 c_5\end{aligned}}{\begin{aligned}&c_2 c_3 + c_2 c_6 + c_3 c_5 + c_5 c_6 + c_2 c_7^2 + c_4 c_6^2 + c_5 c_7^2 + c_6^2 c_7 + c_6 c_7^3 + c_2 c_4 c_7 \\ &+ \; c_3 c_4 c_6 + c_3 c_6 c_7 + c_4 c_5 c_7 + 2c_4 c_6 c_7^2 + c_4^2 c_6 c_7\end{aligned}}$$

**Example 24.** The following is a model from biochemical reaction network theory for the mitogen-activated protein kinase (MAPK) pathway [41] which is part of a molecular signaling network that governs the growth, proliferation, differentiation, and survival of many cell types:

$$\dot{KS_{00}} = -a_{00}KS_{00} + b_{00}KS_{00} + \gamma_{0100}FS_{01} + \gamma_{1000}FS_{10} + \gamma_{1100}FS_{11}$$

$$\dot{KS_{01}} = -a_{01}KS_{01} + b_{01}KS_{01} + c_{0001}KS_{00} - \alpha_{01}FS_{01} + \beta_{01}FS_{01} + \gamma_{1101}FS_{11}$$

$$\dot{KS_{10}} = -a_{10}KS_{10} + b_{10}KS_{10} + c_{0010}KS_{00} - \alpha_{10}FS_{10} + \beta_{10}FS_{10} + \gamma_{1110}FS_{11}$$

$$\dot{FS_{01}} = -\alpha_{11}FS_{11} + \beta_{11}FS_{11} + c_{0111}KS_{01} + c_{1011}KS_{10} + c_{0011}KS_{00}$$

$$\dot{FS_{10}} = a_{00}KS_{00} - (b_{00} + c_{0001} + c_{0010} + c_{0011})KS_{00}$$

$$\dot{FS_{11}} = a_{01}KS_{01} - (b_{01} + c_{0111})KS_{01}$$

$$\dot{K} = a_{10}KS_{10} - (b_{10} + c_{1011})KS_{10}$$

$$\dot{F} = \alpha_{01}FS_{01} - (\beta_{01} + \gamma_{0100})FS_{01}$$

$$\dot{S_{00}} = \alpha_{10}FS_{10} - (\beta_{10} + \gamma_{1000})FS_{10}$$

$$\dot{S_{01}} = \alpha_{11}FS_{11} - (\beta_{11} + \gamma_{1101} + \gamma_{1110} + \gamma_{1100})FS_{11}$$

$$\dot{S_{10}} = -a_{00}KS_{00} + (b_{00} + c_{0001} + c_{0010} + c_{0011})KS_{00} - a_{01}KS_{01}$$
$$+ \; (b_{01} + c_{0111})KS_{01} - a_{10}KS_{10} + (b_{10} + c_{1011})KS_{10}$$

$$\dot{S_{11}} = -\alpha_{01}FS_{01} + (\beta_{01} + \gamma_{0100})FS_{01} - \alpha_{10}FS_{10} + (\beta_{10} + \gamma_{1000})FS_{10}$$
$$- \; \alpha_{11}FS_{11} + (\beta_{11} + \gamma_{1101} + \gamma_{1110} + \gamma_{1100})FS_{11}.$$

This model has 12 state variables

$$KS_{00}(t), \quad KS_{01}(t), \quad KS_{10}(t), \quad FS_{01}(t), \quad FS_{10}(t), \quad FS_{11}(t),$$

$$K(t), \quad F(t), \quad S_{00}(t), \quad S_{01}(t), \quad S_{10}(t), \quad S_{11}(t)$$

and 22 parameters

$$a_{00}, \quad a_{01}, \quad a_{10}, \quad b_{00}, \quad b_{01}, \quad b_{10}, \quad c_{0001}, \quad c_{0010}, \quad c_{0011}, \quad c_{0111}, \quad c_{1011},$$

$$\alpha_{01}, \quad \alpha_{10}, \quad \alpha_{11}, \quad \beta_{01}, \quad \beta_{10}, \quad \beta_{11}, \quad \gamma_{0100}, \quad \gamma_{1000}, \quad \gamma_{1100}, \quad \gamma_{1101}, \quad \gamma_{1110}.$$

We will consider several different cases of what is measured as output. In all of our examples, we attempted to first compute input-output equations using differential elimination via the command *RosenfeldGroebner* in `Maple` [42]. In all of our attempts, the differential elimination failed to terminate meaning that we will just utilize the model equations in the following.

First, for taking the standard 6 measurable outputs:

$$y_1 = K, \quad y_2 = F, \quad y_3 = S_{00}, \quad y_4 = S_{01}, \quad y_5 = S_{10}, \quad y_6 = S_{11},$$

Table 7, computed in about a minute on a single processor, shows that the resulting model is identifiable.

For comparison of methods, neither DAISY [1, 24] nor COMBOS [30] finished the identifiability computations for this model after running for 24 hours. To the best of our knowledge, this is the first successful implementation of a structural identifiability test for this model.

Second, if we adjust the model so that we only take the following 2 measurable outputs:

$$y_1 = K, \quad y_2 = F,$$

Table 8 shows that the resulting model is still identifiable.

Third, if we take the following 4 measurable outputs:

$$y_1 = S_{00}, \quad y_2 = S_{01}, \quad y_3 = S_{10}, \quad y_4 = S_{11},$$

Table 9 shows that the resulting model is still identifiable.

Finally, we consider 10 new mixing parameters, namely

$$ms_{00}, \quad mks_{00}, \quad ms_{01}, \quad mks_{01}, \quad mfs_{01}, \quad ms_{10}, \quad mks_{10}, \quad mfs_{10}, \quad ms_{11}, \quad mfs_{11},$$

**Table 7. Summary of computations showing model is identifiable.**

| $r$ | $\mathrm{corank}_0 \, JF_r$ | $\mathrm{corank}_{22} \, JF_r$ | $d_r$ |
|---|---|---|---|
| 0 | 28 | 6 | 22 |
| 1 | 23 | 1 | 22 |
| 2 | 18 | 0 | 18 |
| 3 | 13 | 0 | 13 |
| 4 | 8 | 0 | 8 |
| 5 | 3 | 0 | 3 |
| 6 | 0 | 0 | 0 |

**Table 8. Summary of computations showing model is identifiable.**

| $r$ | corank$_0$ $JF_r$ | corank$_{22}$ $JF_r$ | $d_r$ |
|---|---|---|---|
| 0 | 32 | 10 | 22 |
| 1 | 30 | 8 | 22 |
| 2 | 28 | 6 | 22 |
| 3 | 26 | 4 | 22 |
| 4 | 24 | 2 | 22 |
| 5 | 22 | 0 | 22 |
| 6 | 20 | 0 | 20 |
| 7 | 18 | 0 | 18 |
| 8 | 16 | 0 | 16 |
| 9 | 14 | 0 | 14 |
| 10 | 12 | 0 | 12 |
| 11 | 10 | 0 | 10 |
| 12 | 8 | 0 | 8 |
| 13 | 6 | 0 | 6 |
| 14 | 4 | 0 | 4 |
| 15 | 2 | 0 | 2 |
| 16 | 0 | 0 | 0 |

with the following 4 measurable outputs:

$$y_1 = ms_{00}S_{00} + mks_{00}KS_{00}$$

$$y_2 = ms_{01}S_{01} + mks_{01}KS_{01} + mfs_{01}FS_{01}$$

$$y_3 = ms_{10}S_{10} + mks_{10}KS_{10} + mfs_{10}FS_{10}$$

$$y_4 = ms_{11}S_{11} + mfs_{11}FS_{11}.$$

Table 10 shows that the resulting model, which has a total of 32 parameters, is unidentifiable with one dimension of unidentifiability.

Using the results above, we can observe from sampling that each irreducible component of a general fiber is simply a line and the following 16 parameters are all identifiable:

$$b_{00}, b_{01}, b_{10}, c_{0001}, c_{0010}, c_{0011}, c_{0111}, c_{1011}, \beta_{01}, \beta_{10}, \beta_{11}, \gamma_{0100}, \gamma_{1000}, \gamma_{1100}, \gamma_{1101}, \gamma_{1110}$$

**Table 9. Summary of computations showing model is identifiable.**

| $r$ | corank$_0$ $JF_r$ | corank$_{22}$ $JF_r$ | $d_r$ |
|---|---|---|---|
| 0 | 30 | 8 | 22 |
| 1 | 26 | 4 | 22 |
| 2 | 22 | 0 | 22 |
| 3 | 18 | 0 | 18 |
| 4 | 14 | 0 | 14 |
| 5 | 10 | 0 | 10 |
| 6 | 6 | 0 | 6 |
| 7 | 2 | 0 | 2 |
| 8 | 0 | 0 | 0 |

**Table 10. Summary of computations showing 1 dimension of unidentifiability.**

| $r$ | $\text{corank}_0\ JF_r$ | $\text{corank}_{32}\ JF_r$ | $d_r$ |
|---|---|---|---|
| 0 | 40 | 8 | 32 |
| 1 | 36 | 4 | 32 |
| 2 | 32 | 0 | 32 |
| 3 | 28 | 0 | 28 |
| 4 | 24 | 0 | 24 |
| 5 | 20 | 0 | 20 |
| 6 | 16 | 0 | 16 |
| 7 | 12 | 0 | 12 |
| 8 | 8 | 0 | 8 |
| 9 | 4 | 0 | 4 |
| 10 | 1 | 0 | 1 |
| 11 | 1 | 0 | 1 |

meaning $a_{00}, a_{01}, a_{10}, \alpha_{01}, \alpha_{10}, \alpha_{11}$ and the 10 mixing parameters are unidentifiable. In fact, no nonconstant linear function in these 16 latter unidentifiable parameters is identifiable.

## Conclusion

In this article, we considered the problems of determining the identifiability of an ODE model, computing the identifiability degree in the case that the model is identifiable and identifiable functions in the case that the model is unidentifiable. To summarize, the results of this article include numerical methods for the following:

1. compute the dimension of unidentifiability with or without input-output equations;

2. for identifiable models, compute the identifiability degree with or without input-output equations using basic homotopy continuation or monodromy loops;

3. for unidentifiable models, compute identifiable and globally identifiable functions inside of a linear family of functions with or without input-output equations.

These methods were illustrated on several examples, including the first known structural identifiability result for MAPK in Example 24.

In the future, we hope to apply similar numerical algebraic geometry methods to other areas in biological modelling, such as controllability, observability, and indistinguishability.

## Acknowledgments

We appreciate the many useful suggestions of the anonymous referees.

## Author Contributions

**Conceptualization:** Daniel J. Bates, Jonathan D. Hauenstein.

**Data curation:** Daniel J. Bates, Jonathan D. Hauenstein, Nicolette Meshkat.

**Formal analysis:** Daniel J. Bates, Jonathan D. Hauenstein, Nicolette Meshkat.

**Investigation:** Daniel J. Bates, Jonathan D. Hauenstein, Nicolette Meshkat.

**Methodology:** Daniel J. Bates, Jonathan D. Hauenstein, Nicolette Meshkat.

**Project administration:** Daniel J. Bates.

**Resources:** Daniel J. Bates, Jonathan D. Hauenstein.

**Software:** Daniel J. Bates, Jonathan D. Hauenstein, Nicolette Meshkat.

**Supervision:** Nicolette Meshkat.

**Validation:** Daniel J. Bates, Jonathan D. Hauenstein, Nicolette Meshkat.

**Visualization:** Jonathan D. Hauenstein, Nicolette Meshkat.

**Writing – original draft:** Daniel J. Bates, Jonathan D. Hauenstein, Nicolette Meshkat.

**Writing – review & editing:** Daniel J. Bates, Jonathan D. Hauenstein, Nicolette Meshkat.

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
