## [Decision Letter · Decision Letter 0]

20 Sep 2019

PONE-D-19-15150

Identifiability and numerical algebraic geometry

PLOS ONE

Dear Prof. Hauenstein,

Thank you for submitting your manuscript to PLOS ONE. After careful consideration, we feel that it has merit but does not fully meet PLOS ONE’s publication criteria as it currently stands. Therefore, we invite you to submit a revised version of the manuscript that addresses the points raised during the review process.

We would appreciate receiving your revised manuscript by Nov 03 2019 11:59PM. To enhance the reproducibility of your results, we recommend that if applicable you deposit your laboratory protocols in protocols.io, where a protocol can be assigned its own identifier (DOI) such that it can be cited independently in the future. For instructions see: http://journals.plos.org/plosone/s/submission-guidelines#loc-laboratory-protocols

We look forward to receiving your revised manuscript.

Kind regards,

Fang-Bao Tian

Academic Editor

PLOS ONE

Journal Requirements:

Additional Editor Comments (if provided):

Thank you for submitting your work to the Plos One. It has been reviewed by two specialists. As you can see, their opinions are quit different. I would encourage you to revise the paper based on their comments, especially those raised by Reviewer 1.

Reviewers' comments:

Reviewer's Responses to Questions

**Comments to the Author**

1. Is the manuscript technically sound, and do the data support the conclusions?

Reviewer #1: Partly

Reviewer #2: Yes

2. Has the statistical analysis been performed appropriately and rigorously? 

Reviewer #1: No

Reviewer #2: Yes

3. Have the authors made all data underlying the findings in their manuscript fully available?

Reviewer #1: Yes

Reviewer #2: Yes

4. Is the manuscript presented in an intelligible fashion and written in standard English?

Reviewer #1: No

Reviewer #2: Yes

5. Review Comments to the Author

Reviewer #1: The paper is well-written and the questions on identifiability posed at the beginning are important. However,

all systems considered in examples have constant coefficients, hence presumably analytical solutions for the main dependent variable are available explicitly in terms of the unknown constant parameters. In reality, systems have non-homogeneous and/or nonlinearities. Can the approach proposed deal with such practical generalities of inhomogeneous and nonlinear systems?

There is some exaggeration of formalism in the first parts of the paper, which is actually not needed.

Can the analysis be extended to PDE's?

The references should be written in the style of the journal and be consistent throughout.

Several references are on arXiv, i.e. not reviewed yet, and limit the credibility of the citations.

The word 'geometry' used is not fully justified and it could be misleading. Suggest removing and find a better formulation.

Given a set of parameters that are desired to estimate and a governing set of equations, can you design the input-output data for identifiability? One can als consult the early works of Kitamura and Nakagiri on the identifiability of distributed systems.

Reviewer #2: I strongly recommend publication after the revisions suggested in the attached PDF. The techniques you provide establish identifiability results for models that would be too large for established methods. As such, it is a substantial contribution to the field of parameter identifiability analysis. Please see attached document for more details.

6. PLOS authors have the option to publish the peer review history of their article (what does this mean?). If published, this will include your full peer review and any attached files.

Reviewer #1: No

Reviewer #2: No

---

## [Author Response · Author response to Decision Letter 0]

31 Oct 2019

Thank you for your careful review. We have incorporated all of your comments into our revised manuscript as described in the cover letter.

---

## [Decision Letter · Decision Letter 1]

25 Nov 2019

Identifiability and numerical algebraic geometry

PONE-D-19-15150R1

Dear Dr. Hauenstein,

We are pleased to inform you that your manuscript has been judged scientifically suitable for publication and will be formally accepted for publication once it complies with all outstanding technical requirements.

With kind regards,

Fang-Bao Tian

Academic Editor

PLOS ONE

Additional Editor Comments (optional):

Reviewers' comments:

Reviewer's Responses to Questions

**Comments to the Author**

1. If the authors have adequately addressed your comments raised in a previous round of review and you feel that this manuscript is now acceptable for publication, you may indicate that here to bypass the “Comments to the Author” section, enter your conflict of interest statement in the “Confidential to Editor” section, and submit your "Accept" recommendation.

Reviewer #2: All comments have been addressed

2. Is the manuscript technically sound, and do the data support the conclusions?

Reviewer #2: Yes

3. Has the statistical analysis been performed appropriately and rigorously? 

Reviewer #2: Yes

4. Have the authors made all data underlying the findings in their manuscript fully available?

Reviewer #2: Yes

5. Is the manuscript presented in an intelligible fashion and written in standard English?

Reviewer #2: Yes

6. Review Comments to the Author

Reviewer #2: (No Response)

7. PLOS authors have the option to publish the peer review history of their article (what does this mean?). If published, this will include your full peer review and any attached files.

Reviewer #2: No

---

## [Editor Report · Acceptance letter]

5 Dec 2019

PONE-D-19-15150R1

Identifiability and numerical algebraic geometry

Dear Dr. Hauenstein:

I am pleased to inform you that your manuscript has been deemed suitable for publication in PLOS ONE. Congratulations! Your manuscript is now with our production department.

With kind regards,

on behalf of

Dr. Fang-Bao Tian

Academic Editor

PLOS ONE